# Innovative In Vivo Imaging and Single Cell Expression from Tumor Bulk and Corpus Callosum Reveal Glioma Stem Cells with Unique Regulatory Programs

**DOI:** 10.3390/cancers17233851

**Published:** 2025-11-30

**Authors:** Natalia dos Santos, Aline Aquino, Friedrich Preußer, Fabio Rojas Rusak, Elisa Helena Farias Jandrey, Miyuki Uno, Tatiane Katsue Furuya, Carmen Lucia Penteado Lancellotti, Marcos Vinicius Calfat Maldaun, Roger Chammas, Stephan Preibisch, Anamaria Aranha Camargo, Cibele Masotti, Erico Tosoni Costa

**Affiliations:** 1Molecular Oncology Center, Hospital Sírio-Libanês, São Paulo 01308-060, SP, Brazil; 2Berlin Institute of Medical Systems Biology, Max Delbrück Centrum for Molecular Medicine, 10115 Berlin, Germany; 3Center for Translational Research in Oncology (LIM24), Instituto do Câncer do Estado de São Paulo (ICESP), Hospital das Clínicas da Faculdade de Medicina da Universidade de São Paulo (HCFMUSP), São Paulo, 01246-000, Brazil; 4Comprehensive Center for Precision Oncology, Universidade de São Paulo, São Paulo, 01246-000, Brazil; 5Departamento de Patologia, Santa Casa de São Paulo, São Paulo 01221-020, SP, Brazil; 6Janelia Research Campus, Howard Hughes Medical Institute, Ashburn, VA 20147, USA

**Keywords:** glioblastoma, glioma stem-like cells, tumor invasion, corpus callosum, intratumoral heterogeneity, single-cell RNA analysis, light-sheet fluorescence microscopy

## Abstract

**Simple Summary:**

Glioblastoma is one of the most aggressive brain tumors because its cells spread widely through the brain, making complete removal impossible. We implanted fluorescently labeled glioma stem-like cells (GSCs) into the mouse brain and observed their infiltrative growth using advanced 3D imaging. Within 6–10 weeks, tumors caused about 15% brain swelling and showed a strong preference for invading the corpus callosum, a major white matter tract. By measuring Sox2-positive cells, we found very high densities in the tumor bulk (over 4000 cells/mm^2^) but much lower levels at the invasive margin (about 30 cells/mm^2^) and a clear gradient in the corpus callosum (3010 to 1.5 cells/mm^2^ across 1 mm). Using CLARITY tissue clearing combined with light-sheet microscopy, we visualized how tumor cells infiltrate along blood vessels and brain structures, adopting distinct shapes and strategies. Single-cell gene analysis of the invading corpus callosum cells identified a 7-gene “invasion signature” (*NES*, *CCND1*, *GUSB*, *NOTCH1*, *E2F1*, *EGFR*, *TGFB1*), which was linked to worse survival and may represent new therapeutic targets.

**Abstract:**

**Background/Objectives**: High-grade gliomas (HGGs), including glioblastomas, are among the most aggressive brain tumors due to their high intratumoral heterogeneity and extensive infiltration. Glioma stem-like cells (GSCs) frequently invade along white matter tracts such as the corpus callosum, but the molecular programs driving this region-specific invasion remain poorly defined. The aim of this study was to identify transcriptional signatures associated with GSC infiltration into the corpus callosum. **Methods**: We established an orthotopic xenograft model by implanting fluorescently labeled human GSCs into nude mouse brains. Tumor growth and invasion patterns were assessed using tissue clearing, light-sheet fluorescence microscopy, and histological analyses. To characterize region-specific molecular profiles, we performed microfluidic-based single-cell RNA expression analysis of 48 invasion- and stemness-related genes in cells isolated from the tumor bulk (TB) and corpus callosum (CC). **Results**: By six weeks post-implantation, GSCs displayed marked tropism for the corpus callosum, with distinct infiltration patterns captured by three-dimensional imaging. Single-cell gene expression profiling revealed significant differences in 7 of the 48 genes (14.6%) between TB- and CC-derived GSCs. These genes—*NES*, *CCND1*, *GUSB*, *NOTCH1*, *E2F1*, *EGFR*, and *TGFB1*—collectively defined a “corpus callosum invasion signature” (CC-Iv). CC-derived cells showed a unimodal, high-expression profile of CC-Iv genes, whereas TB cells exhibited bimodal distributions, suggesting heterogeneous transcriptional states. Importantly, higher CC-Iv expression correlated with worse survival in patients with low-grade gliomas. **Conclusions**: This multimodal approach identified a corpus callosum-specific invasion signature in glioma stem-like cells, revealing how local microenvironmental cues shape transcriptional reprogramming during infiltration. These findings provide new insights into the spatial heterogeneity of gliomas and highlight potential molecular targets for therapies designed to limit tumor spread through white matter tracts.

## 1. Introduction

High-grade gliomas (HGG), including grade 4 IDH-mutant astrocytomas and IDH-wildtype GBMs, are among the most aggressive and heterogeneous primary brain tumors, associated with dismal prognosis [1,2]. A major challenge in their treatment lies in the dual biological features of these tumors: intense angiogenesis within the densely cellular tumor core and diffuse infiltration of glioma stem-like cells (GSCs) along white matter tracts in the peripheral tumor margins. These processes not only drive tumor progression but also limit the efficacy of surgical resection, radiotherapy, and systemic therapies [2]. While both angiogenesis and infiltration have been independently characterized, their interplay and contribution to intratumoral heterogeneity (ITH) remain incompletely understood. A deeper understanding of these spatially distinct yet interconnected processes is essential for improving therapeutic strategies in malignant astrocytomas.

The infiltrative tumor margins, often poorly defined on conventional imaging and histopathology, harbor migrating GSCs capable of escaping treatment and initiating recurrence. Advances in transcriptomic profiling have revealed the extensive transcriptional diversity of tumor cells within HGGs [2,3,4,5]. In a pivotal study, Darmanis et al. [6] used single-cell RNA sequencing (scRNA-seq) to characterize thousands of GBM cells from both the tumor core and invasive periphery, revealing distinct transcriptional signatures that highlighted regional heterogeneity and the invasive potential of peripheral neoplastic cells. Their findings revealed distinct transcriptional differences between these regions, highlighting the heterogeneous nature of GBM and providing insights into the molecular characteristics of infiltrating neoplastic cells at the migrating front of the tumor. Similarly, Patel et al. [2] identified gene meta-signatures in GSCs associated with therapy resistance and tumorigenicity. Notably, the proportion of GSCs ranged from 1% to 22% across tumors, and their expression profiles diverged from hypoxia-associated signatures, despite relatively uniform proliferative indices (Ki67+ staining). These findings, contrasting sharply with in vitro GBM models dominated by proliferative phenotypes, underscore the critical role of the tumor microenvironment in shaping GSC gene expression and functional states.

From a metabolic perspective, Guyon et al. [4] demonstrated that lactate dehydrogenases A and B (LDHA and LDHB) are differentially expressed across GBM regions, facilitating a metabolic symbiosis between the glycolytic tumor core and the more oxygenated invasive margins. Lactate produced in the hypoxic core is metabolized by peripheral cells via oxidative phosphorylation, thereby supporting tumor expansion and infiltration. Dual targeting of LDHA and LDHB disrupted this metabolic interplay, reducing tumor growth and enhancing radiotherapy sensitivity in preclinical models. Together, these studies reveal that glioma cell gene expression is highly dynamic and influenced by spatial context, particularly microenvironmental factors such as vascular proximity and white matter tract association. A comprehensive understanding of how regional microenvironments regulate GSC plasticity, invasion, and therapy resistance is crucial for developing more effective and spatially targeted interventions.

In this study, we orthotopically transplanted fluorescently labeled human GSCs into the brains of Balb/c nude mice, recapitulating the native GBM niche. Following tumor establishment, we employed a novel multimodal approach that integrates tissue-clearing with light-sheet fluorescence microscopy and microfluidic-assisted single-cell isolation for RNA expression analysis. This enabled high-resolution imaging of human GBM cells in optically cleared mouse brains, allowing precise mapping of GSC invasion patterns at single-cell resolution. We compared the expression of 48 genes associated with stemness and aggressiveness across two anatomically distinct brain niches, the corpus callosum (CC) and the tumor bulk (TB). By examining these regions, we assessed how gene expression varies spatially, providing insights into the molecular mechanisms, shaped by the local microenvironment, that drive GSC reprogramming and invasion in vivo. Our findings further suggest that the transcriptional adaptations observed in infiltrating GSCs may converge with molecular programs previously associated with glioblastoma subtypes, including the mesenchymal phenotype described by Phillips et al. (2006) and Verhaak et al. (2010) [7,8]. This context supports the notion that microenvironmental cues can promote transcriptional plasticity, bridging spatial heterogeneity and molecular diversity in HGG.

## 2. Materials and Methods

### 2.1. Cell Lines and Culture Conditions

The human glioblastoma stem-like cell line GSC23 was kindly provided by Dr. Frank Furnari (Ludwig Institute for Cancer Research, University of California, Berkeley). GSC23 cells were cultured as neurospheres in serum-free DMEM/F12 medium supplemented with 2% B27 (Invitrogen), 20 ng/mL epidermal growth factor (EGF; Sigma-Aldrich), 20 ng/mL basic fibroblast growth factor (FGF-2; Sigma-Aldrich), 5 μg/mL heparin (StemCell Technologies), and 1% L-glutamine at 37 °C in a humidified incubator with 5% CO_2_. Cells were routinely tested for mycoplasma contamination. For in vitro sorting and in vivo tracking, GSCs were transduced to stably express the red fluorescent protein Katushka (TurboFP635; Evrogen) as previously described [9].

### 2.2. Animal Studies

All procedures were approved by the Ethics Committee on Animal Use of the Faculty of Medicine, University of São Paulo (CEUA-163/15) and conducted according to institutional guidelines. Immunodeficient Balb/c Nude mice (6–8 weeks old, male) were anesthetized with ketamine (100 mg/kg) and xylazine (10 mg/kg). A sagittal incision exposed the skull, and a small hole was drilled 2 mm lateral and 1 mm anterior to the bregma. A total of 1 × 10^5^ GSC23 cells in 3 μL medium were stereotactically injected into the right striatum at a depth of 3 mm. Postoperative analgesia was administered with tramadol (30 mg/kg, subcutaneously) every 24 h for 2 days. Animals were monitored daily for neurological deficits or weight loss and euthanized at 6 weeks post-injection using isoflurane inhalation (4–5 vol.%) followed by decapitation. This timepoint selected based on prior kinetic analyses of the same GSC23 xenograft model [10]. In that study, longitudinal IVIS Spectrum In Vivo Imaging System (PerkinElmer, Waltham, MA, USA) demonstrated reproducible CC infiltration at six weeks, while avoiding extensive necrosis or distortion of brain architecture that occur at later stages.

### 2.3. Tissue Clearing and Optical Imaging

Whole mouse brains were processed using the CLARITY tissue-clearing protocol [11]. Briefly, freshly collected brains were incubated in a hydrogel monomer solution at 4 °C for 4 days to allow polymer infiltration, followed by thermal polymerization at 37 °C for 3 h. Lipid removal was performed using an SDS-based clearing buffer (4% SDS in 200 mM boric acid, pH 8.5) combined with active electrophoretic clearing in the X-CLARITY system (Logos Biosystems, Anyang, South Korea) at 37 °C for 4 days. After clearing, samples were incubated overnight in a refractive index (RI) matching solution (RI = 1.46, LifeCanvas Technologies, Cambridge, MA, USA) to achieve optical homogeneity. Cleared brains were imaged using two complementary light-sheet fluorescence microscopy (LSFM) systems: a Zeiss Lightsheet Z.1 microscope and a prototype mesoSPIM (Carl Zeiss Microscopy GmbH, Jena, Germany) (mesoscale selective plane illumination microscopy) instrument [12]. Samples were mounted onto the specimen holder with cyanoacrylate-based adhesive and immersed in the imaging chamber filled with the RI-matching solution. Images were acquired using a Zeiss EC Plan-NEOFLUAR 5×/NA 0.16 detection objective and LSFM 5×/NA 0.1 illumination objectives with dual-sided illumination. Two sCMOS cameras (1920 × 1920 pixels, Andor Technology Ltd., Belfast, UK) were used for simultaneous data acquisition. Image stacks were recorded in both Zeiss CZI and RAW formats. For post-acquisition processing, tiled image datasets were registered and fused into single three-dimensional volumes using the BigStitcher software package (version 1.2; Max Planck Institute of Molecular Cell Biology and Genetics, Dresden, Germany) [13]. This workflow enabled seamless reconstruction of large-scale, high-resolution 3D maps of tumor infiltration across the entire mouse brain.

### 2.4. Brain Dissection and GSC Isolation

Following euthanasia, brains were sagittally bisected, and the tumor bulk (TB) and contralateral corpus callosum (CC) were microdissected as described by Collins et al., 2018 [14]. Tissue fragments (<1 mm^3^) were enzymatically dissociated using papain (30 U/mL, 35 °C, 60 min; Worthington Biochemical Corporation, Lakewood, NJ, USA) and subsequently washed in a protease inhibitor solution (Complete™ Mini, Roche Diagnostics, Mannheim, Germany). The resulting cell suspensions were filtered through a 100 μm cell strainer (Corning Inc., Corning, NY, USA) and cultured in serum-free DMEM/F12 medium (Gibco, Thermo Fisher Scientific, Waltham, MA, USA) supplemented as described above, with 1 µg/mL blasticidin (InvivoGen, San Diego, CA, USA). Fluorescent-positive cells were then isolated by fluorescence-activated cell sorting (FACS) using a BD FACSMelody™ cell sorter (BD Biosciences, San Jose, CA, USA).

### 2.5. Histology

Brains not used for ex vivo cultures were fixed in 4% formaldehyde, paraffin-embedded, and sectioned at 10 μm. Sections were deparaffinized, rehydrated, and stained with hematoxylin and eosin (H&E) for histopathological analysis. Micrographs were captured using standard bright-field microscopy.

### 2.6. Single-Cell Gene Expression

A custom panel of 48 target genes and 3 external RNA spike-in controls (ArrayControl™, Thermo Fisher Scientific, Waltham, MA, USA) was designed for single-cell RT-qPCR. The 48-gene panel was curated from an initial dataset of the 1000 most highly expressed transcripts identified by bulk RNA-seq of the GSC23 line cultured in vitro. From this list, five housekeeping genes and forty-three target genes were selected based on their functional relevance to glioma biology, particularly pathways involved in invasion (*VIM*, *ZEB1*, *ZEB2*, *FSCN1*, *CDH2*, *SPARC*, *FN1*, *LOX*), stemness and lineage plasticity (*SOX2*, *NES*, *GFAP*, *PROM1*, *FUS*, *YAP1*, *NOTCH1*), and cell cycle regulation (*CDKN2A*, *CCND1*, *E2F1*, *MCM2*, *TOP2A*, *KIF2C*, *TP53*). Gene prioritization was informed by literature curation and Gene Ontology (GO) annotation to capture representative molecular programs of glioma aggressiveness and stem-like behavior. While additional relevant genes were identified in the broader RNA-seq dataset, their inclusion was restricted by the 96.96 Dynamic Array IFC chip format (Fluidigm, South San Francisco, CA, USA), which limits the number of parallel amplification targets per run. Consequently, the panel was optimized as a focused, yet comprehensive set designed to probe transcriptional heterogeneity across invasion-related pathways. Single-cell capture, lysis, and cDNA synthesis were performed using the Fluidigm C1 system [15], and pre-amplified cDNA was analyzed on 96.96 Dynamic Array IFC chips with the Biomark HD System (Fluidigm, South San Francisco, CA, USA). Quality control was applied at multiple levels to ensure the accuracy and reproducibility of the gene expression data. Cell viability and single-cell integrity were first confirmed using the LIVE/DEAD^®^ Viability/Cytotoxicity Kit (Thermo Fisher Scientific, Waltham, MA, USA), ensuring that only viable and morphologically unique GSCs were analyzed. From the initial pool, 87 CC-derived (91%) and 77 TB-derived (80%) viable single cells were recovered. At the assay level, each qPCR run included internal and external controls to ensure amplification consistency and detect potential artifacts: a No Template Control (NTC) to monitor contamination or nonspecific amplification, a positive control sample to verify reaction efficiency, and an interplate calibrator (IPC) to normalize amplification efficiency across plates. Amplification quality was further verified using the SINGuLAR™ Analysis Toolset 3.0 (Fluidigm, South San Francisco, CA, USA), which automatically identifies reactions with irregular amplification kinetics. Outlier detection based on principal component analysis (PCA) and amplification metrics led to the exclusion of 52 low-quality or inconsistent cells (27%), resulting in 79 CC-derived and 61 TB-derived single GSCs retained for downstream analysis. Expression values represent normalized mean log_2_ expression data averaged from technical replicates across the 48-gene panel. PCA, hierarchical clustering, and violin plots were generated to assess heterogeneity between GSC-TB and GSC-CC populations. The 48-gene panel was derived from RNA-seq data of GSC23 cells cultured in vitro, minimizing immune-related confounding effects and ensuring that the identified CC-Iv signature reflects glioma-cell–specific transcriptional dynamics.

### 2.7. Survival Analysis

Gene expression correlations and survival analyses were performed using GEPIA2 (https://gepia.cancer-pku.cn/, accessed on 24 August 2025) integrating TCGA and GTEx datasets. Patients were stratified by quartiles of CC-Iv signature expression. Kaplan–Meier curves were generated, and statistical significance was determined using log-rank tests.

### 2.8. Data Availability

All single-cell RT-qPCR datasets and imaging data generated in this study will be deposited in publicly available repositories prior to publication. Accession numbers and data links will be provided upon manuscript acceptance. Raw data and analysis code are available from the corresponding author upon reasonable request.

### 2.9. Statistical Analysis

Statistical tests were performed using GraphPad Prism v.9.0.2, with significance defined as *p* < 0.05. All experiments included biological replicates as indicated in figure legends. Data are presented as mean ± standard deviation unless otherwise noted. Multiple-testing corrections (FDR) were applied to the seven CC-Iv genes, with adjusted *p*-values.

## 3. Results

### 3.1. Establishment and Characterization of an Invasive GBM Mouse Model

To generate and characterize a HGG model, we used the glioma stem cell line model GSC23, previously shown to exhibit high levels of Sox2 and a strong tropism for white matter tracts [10]. These cells were genetically modified to stably express the red fluorescent protein TurboFP635 (referred to hereafter as GSC-FP) and were stereotactically injected into the right striatum of BALB/c Nude mice. By 10 weeks post-injection (wkpi), GSC-FP cells had formed large, hypercellular hemispheric tumors characterized by hyperchromatic nuclei, frequent mitoses, and poorly defined margins with extensive satellitosis and infiltration into adjacent neural structures (Figure 1A). The tumor mass exerted a pronounced mass effect, displacing midline structures and ventricles. Consistent with the infiltrative behavior observed in human GBM, GSC-FP cells displayed a strong affinity for the corpus callosum (CC), which appeared both enlarged and distorted due to extensive cellular infiltration (Figure 1A, inserts 1 and 3). In contrast, the anatomically distant anterior commissure remained unaffected (Figure 1A, insert 2), underscoring the regional selectivity of GSC invasion. Based on tumor growth kinetics (Figure 1B), we identified a latency phase followed by a marked acceleration in tumor expansion beginning around 5–6 wkpi. By 10 wkpi, tumor burden resulted in approximately 15% brain swelling (Figure 1C). To focus on earlier stages of tumor invasion and minimize confounding effects from fully developed tumors and associated edema, we selected 6 wkpi as experimental time point. Mice were randomly euthanized at these intervals for downstream three-dimensional imaging and RNA expression profiling, allowing us to capture spatial features prior to the onset of substantial mass effect.

To assess the spatial distribution of GSC-FP cells in the murine brain, we stained coronal brain sections with anti-human Sox2, a transcription factor highly expressed in GSC-FP [10] (Figure 1D). Sox2^+^ cell density (cells/mm^2^) was quantified across three anatomically defined regions: (i) Tumor bulk (TB): the central, densely cellular core of the tumor; (ii) Invasive margins, subdivided into the inner invasive margin (IIM), and the outer invasive margin (OIM) (Figure 1D); (iii) Contralateral corpus callosum (CC), a 1 mm-wide region extending from the longitudinal fissure into the CC of the hemisphere opposite to the tumor (Figure 1E). As shown in Figure 1E, each region was analyzed relative to a defined reference point (black dots in the image). From each point, two concentric zones were defined: an inner zone extending up to 500 μm from the tumor border, and an outer zone extending beyond 1 mm. This approach enabled spatial quantification of GSC-FP/Sox2^+^ cells from the core of each region outward. As shown in Figure 1F, we observed marked spatial heterogeneity in Sox2^+^ cell distribution. As expected, the TB exhibited the highest densities, ranging from 4013 cells/mm^2^ at the central point to 3211 cells/mm^2^ at 1 mm. In the invasive margin, cell density dropped substantially, from 32 cells/mm^2^ at the IIM to 0 cells/mm^2^ beyond 1 mm (OIM). While TB are increasingly dense with GSC cells, at its IIM we noticed a predominant association of GSC-FP with preexisting brain microvasculature (perivascular invasion) (Figure 1A, insert 4). In the contralateral CC, a gradient was also observed, with a density of 3010 cells/mm^2^ at the initial point near longitudinal fissure, which decreased sharply to 1.5 cells/mm^2^ at 1 mm. Sox2^+^ cells in CC were primarily detected as isolated cells or small clusters within the first 500 μm from the initial point, with very few cells found beyond this distance (Figure 1F, corpus callosum).

### 3.2. Complete Mouse Brain Clearing and Tumor Visualization in an Intact, Highly Transparent Brain

To overcome the challenge of imaging whole mouse brains hindered by tissue opacity, we applied the CLARITY tissue-clearing protocol originally developed by Chung et al. (2013) [11]. This method couples hydrogel-tissue hybridization with electrophoretic lipid clearing to render intact brain tissue optically transparent while preserving molecular integrity and structural architecture. Notably, CLARITY is compatible with fluorescent proteins, including the red FP expressed by GSC tumor cells. Brains were initially fixed in PBS, then immersed in a hydrogel monomer solution. During the Hydrogel Monomer Infusion (HMI) step, formaldehyde crosslinked tissue components and covalently tethered monomers to proteins, nucleic acids, and small molecules. Polymerization was then thermally initiated during the Hydrogel-Tissue Hybridization (HTH) step, forming a stable hydrogel-tissue mesh. Lipids and unbound biomolecules were actively removed using Electrophoretic Tissue Clearing (ETC), which enhanced detergent penetration via ionic micelle transport. To further reduce light scattering, samples underwent Refractive Index Matching (RIM) with FocusClear solution, improving optical transparency (Figure 2A). The effectiveness of the CLARITY protocol in producing transparent whole-brain specimens is evident (Figure 2B). To evaluate structural preservation, fluorescence retention, and potential tissue deformation, we imaged cleared brains using light-sheet fluorescence microscopy (LSFM). A Nissl-stained axial view from The Mouse Brain Library (www.mbl.org, accessed on 20 December 2024) was used as an anatomical reference (Figure 2C, left). Autofluorescence in the cleared brain allowed for identification of key structures such as the corpus callosum (CC), lateral ventricle (LV), third ventricle (3V), olfactory bulb (OB), and hippocampus (HC), confirming preserved anatomy. Additionally, strong red fluorescent signals from GSC-FP tumor cells demonstrated compatibility of CLARITY with fluorophores and indicated that tumor architecture was maintained (Figure 2C, right).

We then tested the combined use of CLARITY and LSFM for deep tissue imaging. Optical sectioning was performed on the intact brain, with 1739 optical slices, which allowed us to evaluate the protocol’s effectiveness for imaging the entire mouse brain. Both tumor fluorescence and tissue structures were fully preserved, and LSFM enabled visualization of deeper brain regions, even at the center, highlighting the successful integration of CLARITY with light-sheet imaging (Figure 2D).

### 3.3. Multimodal Invasion Patterns Captured by 3D Imaging of Cleared Brains

To investigate the spatial relationship between glioma cells and the cerebral microvasculature, we employed 3D imaging of cleared brains bearing GSC-FP tumors. Given that brain microvessels typically consist of a central lumen lined by endothelial cells, we sought to characterize how glioma cells interact with these structures during invasion. Classical H&E staining (Figure 3A) revealed hallmark features of Scherer’s secondary structures, including perivascular satellitosis, where tumor cells concentrically surround blood vessels. To validate these observations in a spatially resolved, volumetric context, we leveraged fluorescence-based 3D z-stack imaging (Figure 3B). This approach enabled direct visualization of GSC-FP cells (endogenously fluorescent) infiltrating the brain parenchyma in relation to lumenized blood vessels. Serial optical sections from superficial to deeper planes demonstrated glioma cells encasing vascular structures in a layered fashion, faithfully recapitulating perivascular invasion in three dimensions. These features were particularly enriched at the so-called “inner invasive margin” (IIM), defined as a 200 μm-thick region immediately adjacent to the tumor bulk, where invasive fronts interface with host tissue, including the subpial surface. Perivascular satellitosis emerged as the predominant invasive pattern within the IIM (Figure 3B), underscoring the anatomical fidelity of our model in reproducing established histopathological features. In addition to perivascular invasion, we consistently observed GSC-FP cells dispersing along other neuroanatomical structures such as the subpial space and cortical layers, supporting the notion that glioma cells leverage multiple anatomical corridors, including vasculature, white matter tracts, and subarachnoid surfaces, for dissemination. Further 3D reconstructions revealed distinct cellular invasion modes at the IIM (Figure 3C). GSC-FP cells adopted either a mesenchymal phenotype (elongated and spindle-like morphology), or an amoeboid phenotype (rounded, spherical appearance). These modes of migration were clearly distinguished within the same tumor and across multiple specimens, suggesting that glioma cells exhibit plasticity in their invasive strategies. Together, these findings highlight the strength of the CLARITY-based 3D brain clearing model in preserving and visualizing the complex invasive behavior of glioma cells, as well as their preferential interactions with the brain microenvironment at single-cell resolution.

### 3.4. Spatially Resolved Single-Cell Gene Expression Reveals Distinct Transcriptional States in GSC Subpopulations

To explore region-specific transcriptional programs of infiltrative glioma cells, we based on 3D imaging of invasive patterns to selectively isolate GSC-FP cells from two distinct anatomical niches: TB and contralateral CC. For spatially resolved single-cell RT-qPCR analysis, we curated a panel of 48 genes, partially derived from a whole-transcriptomic screen of over 1000 highly expressed GSC-specific genes (Figure 4A and Appendix A), ensuring relevance to glioma invasion (*VIM*, *ZEB1/2*, *FSCN1*, *CDH2*, *SPARC*, *FN1*, *LOX*), stemness (*SOX2*, *NES*, *GFAP*, *PROM1*, *FUS*, *YAP1*, *NOTCH1*), and cell cycle regulation (*CDKN2A*, *CCND1*, *E2F1*, *MCM2*, *TOP2A*, *KIF2C*, *TP53*). Additionally, we incorporated three interplate calibrators (synthetic RNA spikes) to ensure quality control across qPCR runs. Using this gene set, we initially analyzed a total of 180 single cells collected from two spatially distinct regions of the mouse brain at 6 weeks post-injection (6 wpi): 90 GSCs from the center of the primary tumor in the left hemisphere (tumor bulk, GSC-TB) and 90 GSCs from the corpus callosum (GSC-CC) of the contralateral hemisphere (Figure 1E, TB and CC initial points). Following two rounds of quality control (first for cell viability and single-cell integrity, and subsequently for amplification consistency and outlier exclusion) 140 high-quality single cells (79 GSC-CC and 61 GSC-TB) were retained for downstream analysis.

To contextualize the regions from which GSC-TB and GSC-CC cells were isolated, we first examined the structural integrity of mouse brains at 6-wkpi. The ipsilateral hemisphere exhibited subtle anatomical changes associated with early tumor progression, including modest reductions in gray matter density and a mild midline shift. These alterations were accompanied by early signs of brain displacement, but without overt necrosis in the TB. In contrast, the contralateral hemisphere, source of GSC-CC cells, appeared largely histologically preserved, showing only a slight deformation of the right lateral ventricle, likely due to the compressive influence of the expanding ipsilateral mass. Notably, these structural changes did not result in overt neurological symptoms or weight loss, confirming the viability of animals at the time of sample collection.

To assess the viability and specificity of GSCs within the mouse brain, we used the LIVE/DEAD^®^ Viability/Cytotoxicity Kit and flow cytometric analysis, gating for calcein-AM^+^ (viable) and FP^+^ (human) cells. After isolating the GSC subpopulations and preparing them as single-cell suspensions, we performed individual cell isolation using an integrated fluidic circuit (IFC) within the Fluidigm C1 system. A total of 180 single cells were initially captured (90 GSC-CC and 90 GSC-TB). After the first quality-control step, 164 viable single GSCs (87 GSC-CC, 77 GSC-TB) were selected. Following a second QC round based on amplification consistency and outlier detection, 140 high-quality single cells (79 GSC-CC and 61 GSC-TB) were retained for downstream expression analysis. Non-viable cells, debris, or non-single cells (doublets/triplets) were excluded.

Principal component analysis (PCA) of these 140 high-quality single GSCs revealed a clear segregation between GSC-CC and GSC-TB subpopulations. The first principal component (PC1) accounted for the greatest variance and effectively separated the two populations into distinct transcriptional groups (Figure 4B). Unsupervised hierarchical clustering complemented this analysis, further stratifying the cells into three clusters (Figure 4C). Cluster 1 consisted predominantly of GSC-CC cells (91%, 32/35) with high expression of 85% (41/48) of the selected genes. Cluster 2 was enriched in GSC-TB cells (90%, 19/21) and displayed uniformly low expression levels. Cluster 3 represented a heterogeneous transitional state, comprising 56% GSC-CC (59/108) and 44% GSC-TB (48/108), with variable gene expression. Together, these analyses demonstrate spatially defined transcriptional states among infiltrative versus tumor bulk–derived glioma stem-like cells, revealing dynamic mRNA alterations associated with cell invasion through the corpus callosum over six weeks post-implantation.

### 3.5. Differential Expression Genes (DEG) Reveal a Corpus Callosum Invasion Signature in Glioma Stem Cells

Next, using the SINGuLAR™ Analysis Toolset 3.0 (*p* < 0.05 and a minimum linear fold change of 20%), we identified seven differentially expressed genes (14.6% of the panel) between TB- and CC-derived GSCs: *NES*, *CCND1*, *GUSB*, *NOTCH1*, *E2F1*, *EGFR*, and *TGFB1* (Table 1). These genes, collectively designated as the Corpus Callosum Invasion signature (CC-Iv signature), exhibited increased expression in GSC-CC cells relative to TB cells. Although the expression differences were subtle, they were statistically significant and reproducible across samples, supporting the relevance of the CC-Iv signature. As expected, five housekeeping genes (*ACTB*, *HMGB1*, *GAPDH*, *GFAP*, and *RPLP0*) showed no significant expression differences, confirming the specificity of the observed variation.

Violin plots of the CC-Iv genes revealed distinct expression distributions between GSC-TB and GSC-CC populations. GSC-TB cells exhibited a defined bimodal expression pattern, suggesting transcriptional heterogeneity (Figure 5A). In contrast, GSC-CCs displayed a broader, elevated distribution with less pronounced bimodality, consistent with a more uniformly activated invasive program. Interestingly, the bimodality observed in GSC-TBs may indicate that a subset of these cells shares molecular features with GSC-CCs. One possibility is that this subpopulation represents tumor cells located near or at the corpus callosum within the ipsilateral hemisphere, cells that may already be engaging invasion-associated pathways due to their proximity to permissive white matter structures.

Next, we performed unsupervised hierarchical clustering of the 7-genes associated with CC-Iv signature. This analysis identified two major clusters: one enriched for GSC-TB cells and another predominantly composed of GSC-CCs (Figure 5B). For instance, Cluster 2, dominated by GSC-CCs, contained only 7.5% (6/79) of GSC-TB cells, reinforcing the notion that anatomical localization contributes to the emergence of distinct transcriptional subtypes.

To explore the functional connectivity of the CC-Iv signature genes, we analyzed predicted protein–protein interactions using STRING [16]. Among the seven encoded proteins, GUSB was the only one lacking predicted interactions with the other CC-Iv members. In contrast, NOTCH1 emerged as a central hub within the network, showing direct and indirect interactions with EGFR, NES, TGFB1, E2F1, and CCND1, forming a dense signaling network centered on NOTCH1 (Figure 5C). The presence of such a cohesive interaction map suggests that the CC-Iv genes may participate in shared or converging signaling pathways. This suggests that NOTCH1 could act as a key integrator of stemness, proliferation, and invasive behaviors in GSC-CC cells, coordinating molecular programs that are spatially restricted to white matter-infiltrating glioma populations.

### 3.6. Confirmation of CC-Iv Signature Co-Expression and Clinical Relevance Using Public GBM Datasets

To determine whether the CC-Iv gene signature reflects a coordinated transcriptional program in human gliomas, we performed gene expression correlation analyses in GBM using the GEPIA2 platform. Strong and statistically significant positive correlations were observed between *NOTCH1* and four other CC-Iv genes: *CCND1* (r = 0.62, *p* = 1.5 × 10^−18^) (Figure 6A), *E2F1* (r = 0.56, *p* = 1.5 × 10^−14^) (Figure 6B), *EGFR* (r = 0.32, *p* = 2.9 × 10^−5^) (Figure 6C), and *NES* (r = 0.63, *p* = 1.7 × 10^−19^) (Figure 6D). These findings suggest that *NOTCH1* may act as a central regulator within this transcriptional program, potentially coordinating processes related to stemness and invasion. In contrast, no significant associations were detected for *TGFB1* or *GUSB*, indicating possible divergent regulatory control. The high degree of co-expression supports the hypothesis that spatially distinct GSC populations in white matter tracts adopt a conserved invasive gene expression program in GBM.

To assess the potential clinical relevance of the CC-Iv signature, we used GEPIA to stratify glioma patients by their aggregate CC-Iv expression score (upper vs. lower quartile). Kaplan–Meier survival analyses revealed that a high CC-Iv signature score was significantly associated with worse overall survival (*p* = 1.9 × 10^−6^) and disease-free progression (*p* = 1.5 × 10^−4^) in patients with LGG, but not in HGG patients (*p* > 0.05) (Figure 6E–H). This differential prognostic impact suggests that while CC-Iv genes may be crucial for glioma progression and infiltration in low-grade disease, their expression in HGG may already be constitutively elevated, reducing their stratification power in advanced cases. Accordingly, an independent analysis using GEPIA2 platform revealed that CC-Iv signature expression is significantly higher in GBM than in LGG (1.7-fold increase, *p* < 0.001), supporting its constitutive activation in high-grade gliomas (Appendix A).

## 4. Discussion

Despite major advances in multimodal therapies, HGGs remain among the most aggressive and intractable human cancers, with recurrence occurring in over 90% of cases, often within 2 cm of the resection margin [17]. This high recurrence rate is partly driven by GSCs, which migrate beyond surgical borders and resist conventional therapies [18]. These infiltrative GSCs therefore represent a central challenge for durable disease control, as they fuel therapeutic failure, tumor progression, and poor prognosis.

In line with recent single-cell and spatial transcriptomic studies, our findings reinforce the concept that gliomas are not homogeneous entities but display profound spatial and transcriptional heterogeneity. The infiltrative margin is not merely a passive extension of the tumor core but a dynamic niche enriched with plastic GSCs that switch phenotypes in response to microenvironmental cues, a process consistent with the “Go or Grow” model [19]. As previously shown, cells in hypoxic and densely cellular cores tend to activate proliferative and angiogenic programs [2,20], while peripheral cells, especially those invading white matter, exhibit migratory, stem-like, and developmental gene expression [6,21].

Using an orthotopic xenotransplantation model combined with advanced imaging and spatial transcriptomics, we captured the complexity of glioma infiltration through the CC, a key anatomic route for contralateral spread. This model reproduced hallmark features of diffuse astrocytomas, including ill-defined margins and preferential invasion along white matter tracts. Importantly, infiltrating GSCs did not expand concentrically but followed substrate-guided migration, generating structurally complex and clinically elusive invasive fronts. These observations provide a mechanistic explanation for the limited efficacy of standard resection margins.

At the molecular level, we identified a spatially restricted transcriptional program in CC-Iv cells, comprising seven genes (*GUSB*, *NOTCH1*, *E2F1*, *EGFR*, *NES*, *CCND1*, and *TGFB1*). This program was enriched for functions linked to migration, stemness, and apoptosis resistance. *NES* and *CCND1* were most robustly upregulated, highlighting their potential relevance. *NES* contributes to cytoskeletal dynamics and motility and has been clinically associated with poor prognosis [22], while *CCND1* regulates the G1/S transition and has been implicated in therapy resistance, mesenchymal transition, and increased motility in GBM [23,24]. *GUSB* may facilitate extracellular matrix remodeling [25], whereas *NOTCH1* upregulation reinforces the role of Notch signaling in sustaining stemness and invasion, potentially via ligand-mediated activation within white matter niches [26].

Analysis of bulk GBM transcriptomic data (GEPIA) further demonstrated significant co-expression of CC-Iv genes in patient tumors, suggesting a coordinated regulatory network underlying peritumoral invasion. High CC-Iv expression correlated with worse survival in LGG but not GBM, likely reflecting constitutively high expression in high-grade disease. This raises the possibility that CC-Iv genes may act as early markers of malignant progression, particularly in lower-grade gliomas.

Validation at the single-cell level using the dataset by Darmanis et al. [6] confirmed partial overlap, with *NOTCH1* and *EGFR* enriched in peripheral GBM cells. The incomplete concordance likely reflects methodological differences: while prior studies sampled broadly from tumor peripheries, our anatomically precise sampling of CC-infiltrating GSCs captured a distinct and clinically relevant subpopulation. This underscores the necessity of spatially resolved profiling to dissect glioma heterogeneity and to develop anatomically informed biomarkers.

Mechanistically, our results position *NOTCH1* at the center of the CC-Iv program. The coordinated expression of *NOTCH1*, *EGFR*, and *NES* suggests a synergistic axis that sustains stemness and motility of CC-invading GSCs. Prior reports have shown that CD133+/*NOTCH1*+ GSCs preferentially localize to white matter tracts, where Jagged ligands reinforce a *NOTCH1*–SOX2 feedback loop to maintain the invasive phenotype [26]. Our findings extend this model by highlighting how demyelinated axonal niches may serve as hubs for Notch-driven invasion across the CC.

Although the present study did not include targeted functional validation of CC-Iv genes, our goal was to establish an anatomically defined proof-of-concept framework integrating in vivo imaging and spatial single-cell profiling. The associations reported here are therefore correlative rather than causal, serving as a foundation for future mechanistic testing. This approach is consistent with current best-practice guidelines for single-cell genomics validation, which recommend phased validation strategies, initially defining spatially restricted transcriptional programs, followed by orthogonal and functional assays in subsequent studies [27].

Whether the CC-Iv transcriptional program represents a static feature (reflecting selective enrichment of GSCs already expressing this signature) or a plastic adaptation induced by the CC microenvironment remains an open question. Observations from ongoing analyses in our laboratory suggest that this expression pattern is lost after cell culture expansion, suggesting that CC-Iv is a context-dependent and reversible state maintained by local microenvironmental cues rather than a fixed genetic trait. Furthermore, the dynamic nature of the CC-Iv signature suggests that therapy-induced state transitions may further shape infiltration phenotypes and emerge as a mechanism of adaptive invasion. Several genes included in our signature, notably *NOTCH1*, *NES*, *E2F1* and *EGFR*, have been implicated in glioma adaptation to temozolomide (TMZ) therapy, where changes in their expression or activity influence stemness, invasion and drug resistance [28,29,30]. These observations reinforce the idea that the CC-Iv signature may not represent a fixed state but a plastic and treatment—responsive transcriptional program, highlighting the need to explore how therapy-induced transitions may further shape infiltration phenotypes.

Interestingly, the CC-Iv signature identified in our work shares substantial overlap with mesenchymal-like gene networks previously associated with invasive glioblastoma subtypes [7,8]. The enrichment of regulators such as *NOTCH1*, *TGFB1*, and *EGFR* suggests that GSCs invading the CC may activate transcriptional modules reminiscent of mesenchymal transition, enabling enhanced motility, metabolic plasticity, and resistance to stress. This mesenchymal-like reprogramming likely reflects an adaptive response to the distinct biochemical and structural properties of white matter tracts, where migratory cues dominate over proliferative signals.

Notably, this interpretation aligns with observations from non-neoplastic systems. Namestnikova et al. (2023) demonstrated that mesenchymal stromal cells transplanted into the contralateral striatum of rats with experimental ischemic stroke migrate through the CC toward the opposite hemisphere, underscoring the permissive nature of this white-matter tract as a cellular migration route [31]. Such parallels raise the intriguing possibility that GSCs may transcriptionally and functionally resemble normal mesenchymal cells that naturally exploit the CC as a migratory path. This convergence supports the idea that the CC-Iv program reflects not only oncogenic adaptation but also the co-option of conserved mesenchymal motility pathways, contributing to the highly invasive and recurrent behavior that characterizes infiltrative gliomas.

## 5. Conclusions

This study defines a novel transcriptional program (CC-Iv) that characterizes GSCs infiltrating the corpus callosum, encompassing seven genes (*GUSB*, *NOTCH1*, *E2F1*, *EGFR*, *NES*, *CCND1*, and *TGFB1*) associated with migration, stemness, and resistance to apoptosis. Among these, *NES* and *CCND1* emerged as particularly relevant to infiltration, integrating cytoskeletal remodeling and cell cycle regulation with invasive phenotypes. By combining orthotopic xenotransplantation, advanced imaging, and spatial transcriptomics, our model faithfully reproduced hallmark features of diffuse glioma infiltration, offering mechanistic insight into how glioma cells exploit white matter tracts to evade conventional therapies.

Notably, the CC-Iv program shares molecular regulators with mesenchymal-like transcriptional networks, suggesting that invading glioma stem-like cells may co-opt conserved migratory pathways similar to those used by mesenchymal cells.

Clinically, CC-Iv genes may serve as early biomarkers of malignant progression, especially in LGG, and provide region-specific therapeutic targets. Mechanistic dissection of this network is a key next step to determine the causal contribution of *NOTCH1*, *EGFR*, and *NES* to CC infiltration. Targeting these nodes, either individually or in rational combinations, may open new avenues for anti-invasion therapies. Ultimately, integration of anatomically precise spatial transcriptomics into translational pipelines could enable the development of personalized strategies that limit peritumoral spread, reduce recurrence, and improve patient outcomes across glioma grades.

Although this study was based on a single, well-characterized GSC line (GSC23) selected for its reproducible tropism to the CC, it provides a proof-of-principle framework to investigate spatially defined transcriptional programs of glioma invasion. Reproducing these analyses across additional, molecularly diverse GSC models will be important to validate the generalizability and biological robustness of the CC-Iv signature. Further validation using complementary models and functional perturbation assays will also be essential to confirm the causal roles of CC-Iv genes in invasion and therapy response. These follow-up efforts will align with current scRNA-seq validation standards, which recommend orthogonal and functional approaches to strengthen mechanistic inference [27]. Notably, partial overlap between the CC-Iv signature and peripheral glioma cell programs described by Darmanis et al. [6] further supports its biological relevance and indicates that similar invasive states may recur across distinct datasets, despite methodological differences in sampling and profiling.

## Figures and Tables

**Figure 1 cancers-17-03851-f001:**
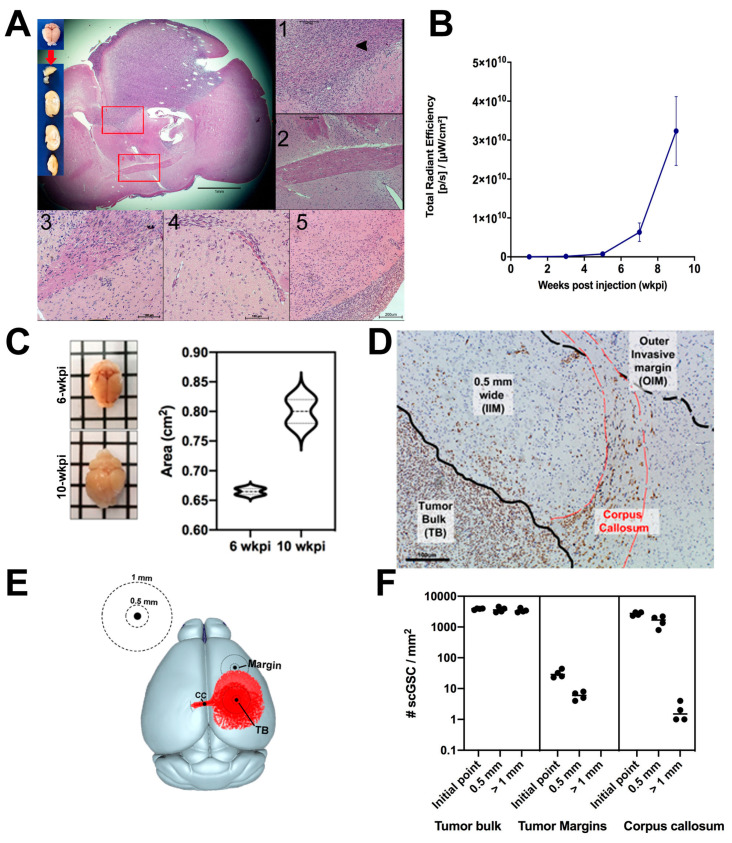
Invasive behavior and spatial distribution of Sox2^+^ glioma stem-like cells (GSCs) in distinct brain regions of GSC23 tumor-bearing mice. (**A**) Macroscopic view (left insets) of a Balb/c Nude mouse brain 10 wkpi with GSC23 glioma stem cells. H&E-stained sections reveal extensive tumor cell infiltration into the brain parenchyma. Insets highlight: (**1**) GSC23 cells infiltrating spinal cord-associated white matter tracts (black arrow, tumor cells invading the corpus callosum on the ipsilateral (tumor-bearing) hemisphere); (**2**) anterior commissure; (**3**) cellular infiltration within the corpus callosum on the contralateral hemisphere; (**4**) perivascular invasion; and (**5**) subpial invasion. Neuropathological analysis was performed in collaboration with a certified neuropathologist. (**B**) Fluorescence-based tumor growth curves of Balb/c Nude mice inoculated with 1.5 × 10^3^ GSC cells into the right striatum (day 0), monitored at 10–14 day intervals using the IVIS Spectrum imaging system. (**C**) Brain area expansion in a GSC-FP xenograft model. **Left**: Representative whole-brain images from mice at 6-wkpi (top) and 10-wkpi (bottom), demonstrating progressive tumor-associated brain enlargement. **Right**: Quantification of total brain area reveals a 15% increase at 10 wkpi compared to 6 wkpi. Data are shown as mean ± SD. Statistical significance was determined using an unpaired *t*-test (*p* = 0.041). (**D**) Spatial distribution of Sox2^+^ GSC-FP cells in a xenograft GBM model. Representative immunohistochemical image of a coronal mouse brain section stained with anti-human Sox2, highlighting three anatomically defined regions used for spatial quantification: (i) Tumor Bulk (TB), outlined in black; (ii) Inner Invasive Margin (IIM), a 0.5 mm peritumoral zone directly adjacent to the tumor border; and (iii) Outer Invasive Margin (OIM), a peripheral zone beyond the IIM, extending further into the white matter. Red lines demarcate the boundaries of the corpus callosum at the ipsilateral hemisphere, into which Sox2^+^ cells can be seen infiltrating. Sox2^+^ cell density was quantified in these regions to assess the spatial dynamics of human GSC-FP invasion. (**E**) Schematic diagram of the inoculation site (TB, black dot) and regions of interest. Concentric circles were drawn at 500 μm and 1 mm radii from the tumor center, tumor margin, and the start of the contralateral corpus callosum. Tumor cells are represented in red. (**F**) Quantification of Sox2-positive human glioma cells in three defined anatomical regions: tumor bulk (TB), tumor margins, and contralateral corpus callosum (CC), based on immunostaining of brain sections. Cell counts were further stratified by distance relative to each region (center or initial point, 500 μm, and beyond 1 mm from center). Data represent five independent tumors (n = 5). A nested ANOVA revealed significant differences in Sox2^+^ cell density across regions and distances (*p* < 0.05), highlighting regional heterogeneity and invasive capacity of GSC-FP cells.

**Figure 2 cancers-17-03851-f002:**
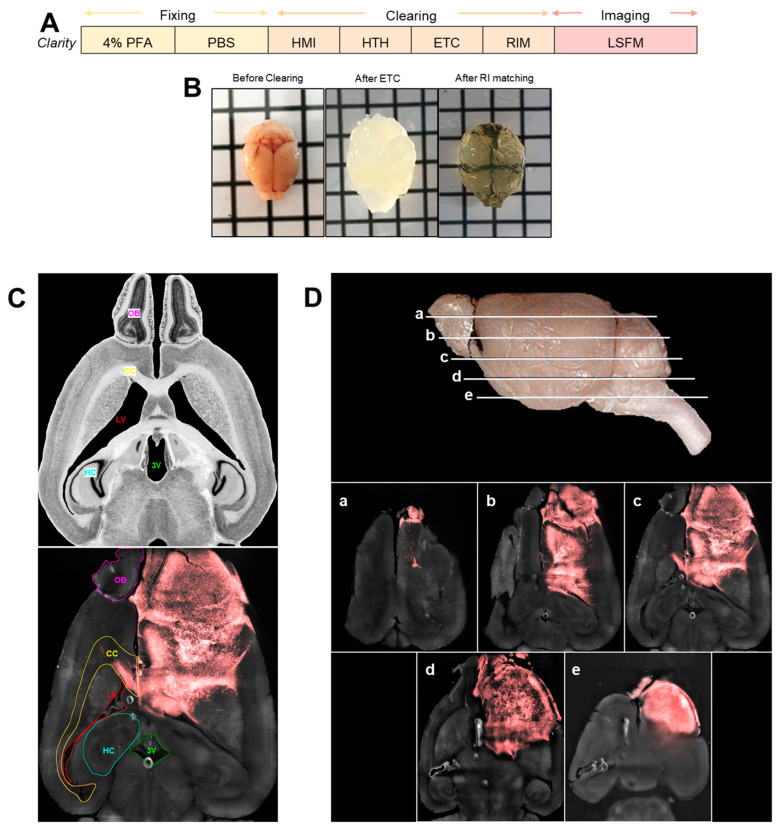
CLARITY protocol and visualization of cleared mouse brains. (**A**) Schematic representation of the CLARITY sample clearing workflow. Mouse brains were fixed, washed, and stored in PBS prior to clearing. During the Hydrogel Monomer Infusion (HMI) step, tissues were incubated in a solution containing formaldehyde and hydrogel monomers. This was followed by Hydrogel-Tissue Hybridization (HTH), where polymerization formed a stable hydrogel-tissue mesh. Brains were then subjected to Electrophoretic Tissue Clearing (ETC) to actively remove lipids, and finally immersed in FocusClear solution for Refractive Index Matching (RIM). Imaging was performed using light sheet fluorescence microscopy (LSFM). (**B**) Macroscopic images of a mouse brain before clearing (left), after ETC (middle), and after RIM (right), illustrating progressive tissue transparency. (**C**) (on left) Nissl-stained axial view of a mouse brain from The Mouse Brain Library (www.mbl.org, accessed on 20 December 2024), used as an anatomical reference. (on right) Representative image of a 10-wkpi mouse brain cleared using the CLARITY protocol and imaged with LSFM at low resolution (on right). Major anatomical structures including the corpus callosum (CC), lateral ventricles (LV), hippocampus (HC), third ventricle (3V), and olfactory bulb (OB) remained intact. Tumor tissue expressing a red fluorescent protein was clearly distinguishable from the autofluorescent background, confirming that fluorescence was preserved throughout the CLARITY procedure. (**D**) (on top) Representative image of the mouse brain (The Mouse Brain Library, www.mbl.org, accessed on 20 December 2024) sectioned in different areas corresponding to the images below (**a**–**e**) demonstrating the efficiency of the combination of the CLARITY protocol and z-stack light sheet microscopy in clearing and imaging the whole mouse brain maintaining the tumor fluorescence and preserving the morphology of the structures. The complete 3D light-sheet microscopy dataset for Figure 2C,D is publicly available at the Janelia Research Campus data repository (https://data.janelia.org/m9otzh, accessed on 27 October 2025).

**Figure 3 cancers-17-03851-f003:**
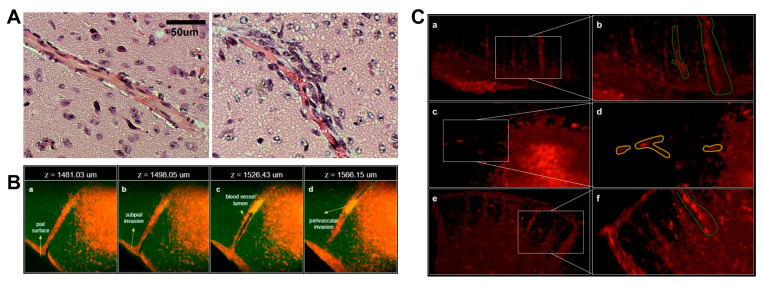
Perivascular invasion and 3D infiltration patterns of glioma cells at single-cell resolution. (**A**) H&E-stained brain sections illustrating vascular structures. Left: normal blood vessel surrounded by non-neoplastic parenchyma. Right: classic perivascular satellitosis, a hallmark of Scherer’s secondary structures, showing glioma cells radially aligned around a central blood vessel. (**B**) High-resolution 3D fluorescence z-stack images of a cleared brain depicting GSC-FP cell invasion at the inner invasive margin (IIM). (**a**) Superficial plane (z = 1481.03 μm); (**b**) subpial invasion; (**c**) infiltration along a lumenized microvessel; (**d**) perivascular invasion (z = 1566.15 μm). This 3D organization recapitulates classical histopathological features in situ and underscores the anatomical fidelity of the model. (**C**) Representative 3D tile-scan reconstructions of GSC-FP invasion at the IIM showing distinct infiltration phenotypes. (**a**) Collective, cluster-based migration (green outlines); (**b**) detailed view of panel (**a**) highlighting cohesive cellular clusters; (**c**) mesenchymal single-cell migration (yellow outlines); (**d**) detailed view of panel (**c**) showing elongated, individually migrating cells; (**e**) infiltration along a lumenized microvessel; (**f**) detailed view of panel (**e**) illustrating perivascular alignment of invasive cells. Insets highlight morphological features that distinguish these invasive patterns within the 3D brain microenvironment. The complete 3D light-sheet microscopy dataset for (**C**) is publicly available at the Janelia Research Campus data repository (https://data.janelia.org/m9otzh, accessed on 27 October 2025).

**Figure 4 cancers-17-03851-f004:**
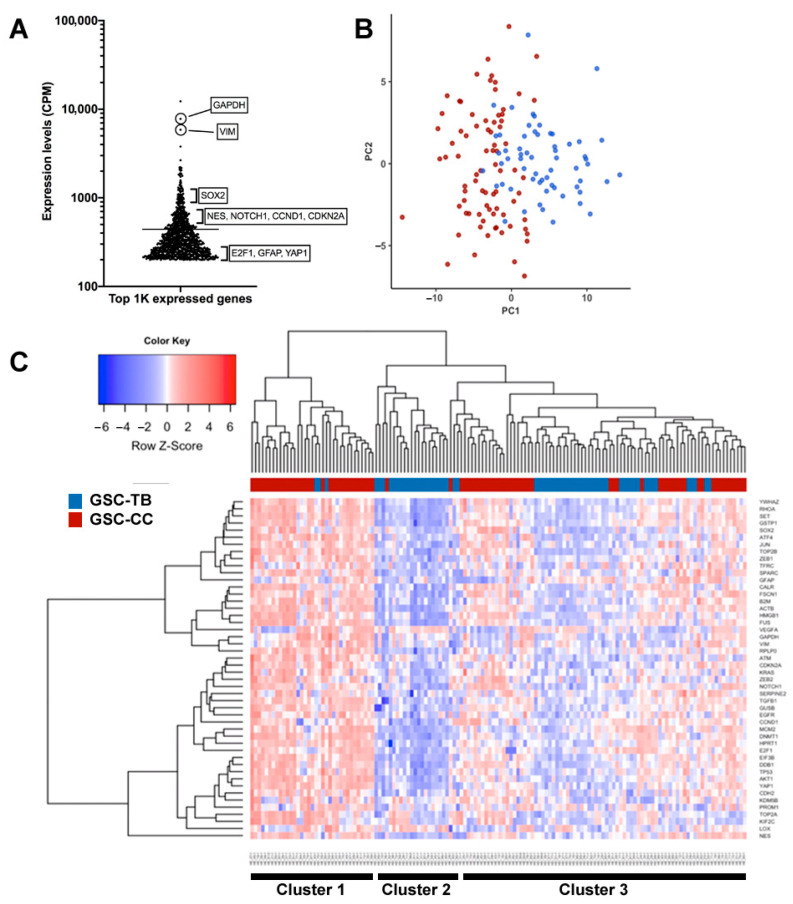
Transcriptional states distinguish GSC-TB and GSC-CC subpopulations. (**A**) Ranked expression levels of the top 1000 most highly expressed genes in GSC-FP cells. Key genes associated with metabolism (*GAPDH*), mesenchymal identity (*VIM*), stemness (*SOX2*, *NES*, *NOTCH1*), and proliferation (*CDKN2A*, *CCND1*, *E2F1*, *YAP1*) are highlighted. (**B**) Principal Component Analysis (PCA) of GSC-TB (blue) and GSC-CC (red) single-cell RT-qPCR data, showing clear separation along PC1, reflecting divergent transcriptional states between the two subpopulations. (**C**) Unsupervised hierarchical clustering of GSC-TB and GSC-CC single cells (n = 140; 79 GSC-CC, 61 GSC-TB) based on the expression of 48 selected genes. Each column represents one cell and each row one gene. Three major clusters emerged: Cluster 1 (predominantly GSC-CC) with uniformly high gene expression, Cluster 2 (mainly GSC-TB) with uniformly low expression, and Cluster 3 (a heterogeneous mix of both cell types) displaying variable expression profiles.

**Figure 5 cancers-17-03851-f005:**
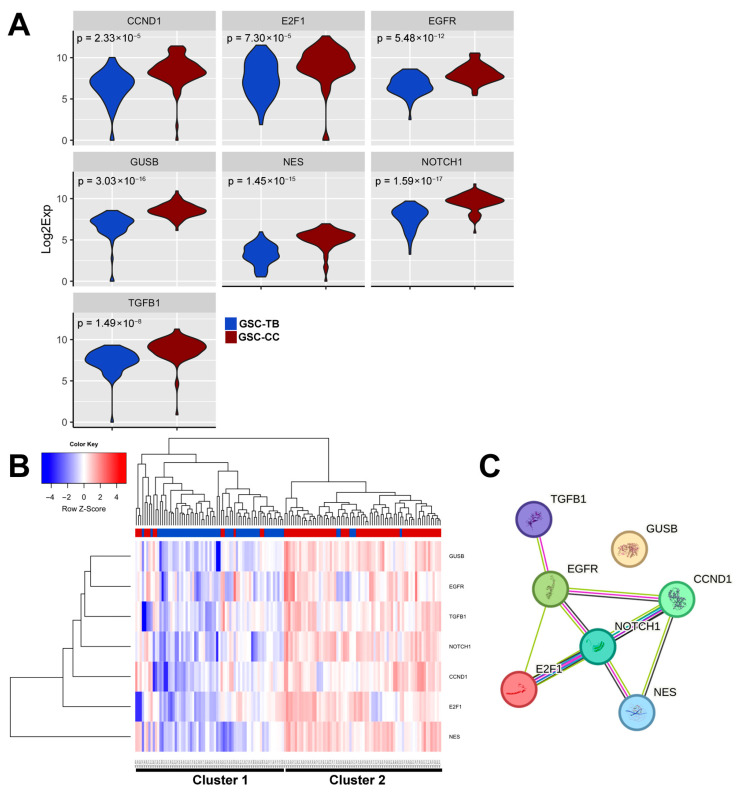
Transcriptional and interaction analysis of the CC-Iv gene signature. (**A**) Violin plots showing the expression distribution of the seven CC-Iv signature genes (*NES*, *CCND1*, *GUSB*, *NOTCH1*, E2F1, EGFR, and *TGFB1*) across single cells. Blue distributions represent GSC-TB cells, while red distributions represent GSC-CC cells. The bimodal pattern in GSC-TB and broader expression range in GSC-CC suggest region-specific transcriptional plasticity. (**B**) Unsupervised hierarchical clustering of scGSCs (n = 164) based on the expression of the seven CC-Iv genes. Columns represent individual cells (77 GSC-TB in blue, 87 GSC-CC in red), and rows represent genes. Analysis was performed using the SINGuLAR™ Analysis Toolset 3.0. Distinct clustering patterns highlight transcriptional differences between the two subpopulations. (**C**) Protein–protein interaction network of the CC-Iv gene products predicted by the String (version 12.0). The network was generated using a minimum interaction confidence score of 0.700 (high confidence). Interaction edges indicate different sources: experimental or known interactions (pink/light blue), gene co-occurrence (dark blue), co-expression (black), and text mining (light green). *NOTCH1* appears as a central hub connecting most other members, except *GUSB*, which is not predicted to interact with the others.

**Figure 6 cancers-17-03851-f006:**
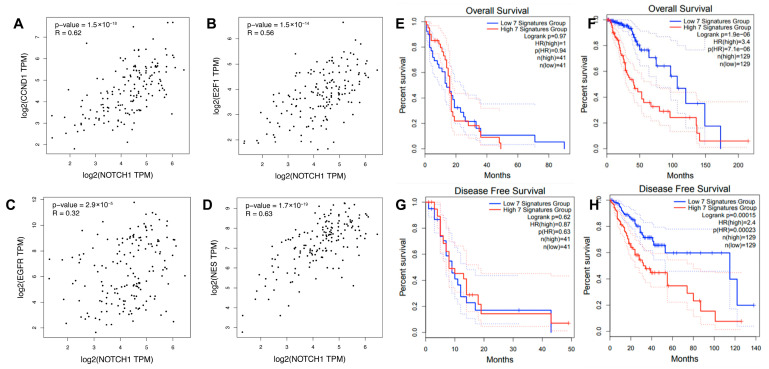
Correlation and prognostic value of CC-Iv signature genes in glioma. (**A**–**D**) Spearman correlation analyses between *NOTCH1* and other CC-Iv signature genes in HGG samples (n = 163), using the GEPIA2 platform (http://gepia2.cancer-pku.cn, accessed on 24 August 2025). Strong and statistically significant positive correlations were observed between *NOTCH1* and *NES* (r = 0.63, *p* = 1.7 × 10^−19^), *CCND1* (r = 0.62, *p* = 1.5 × 10^−18^), *E2F1* (r = 0.56, *p* = 1.5 × 10^−14^), and EGFR (r = 0.32, *p* = 2.9 × 10^−5^), suggesting co-regulation within a shared transcriptional network. No significant correlations were found for *TGFB1* (r = 0.09, *p* = 0.26) and *GUSB* (r = 0.16, *p* = 0.04). (**E**–**H**) Kaplan–Meier survival curves depicting overall survival (OS) and disease-free survival (DFS) in glioma patients stratified by the median expression of the seven CC-Iv signature genes. Analyses were performed for HGG ((**E**,**G**); n = 82) and LGG ((**F**,**H**); WHO grades II and III; n = 258).

**Table 1 cancers-17-03851-t001:** Seven-gene of the Corpus Callosum Invasion signature.

Gene Symbol	Gene Name	HGNC ID	Fold-Change	FDR
*NES*	Nestin	HGNC:7757	1.6	1.02 × 10^−14^
*CCND1*	Cyclin D1	HGNC:1582	1.3	1.70 × 10^−7^
*GUSB*	Glucuronidase Beta	HGNC:4696	1.3	2.12 × 10^−15^
*NOTCH1*	Notch Receptor 1	HGNC:7881	1.25	1.11 × 10^−16^
*E2F1*	E2F Transcription Factor 1	HGNC:3113	1.21	5.11 × 10^−4^
EGFR	Epidermal Growth Factor Receptor	HGNC:3236	1.2	3.84 × 10^−11^
*TGFB1*	Transforming Growth Factor Beta 1	HGNC:11766	1.2	1.04 × 10^−7^

## Data Availability

The raw 3D light-sheet microscopy datasets supporting Figure 2C,D and Figure 3C are publicly available at the Janelia Research Campus data repository: https://data.janelia.org/m9otzh (accessed on 24 October 2025). All other data supporting the findings of this study are available from the corresponding author upon reasonable request.

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
