# Peer review of "Innovative In Vivo Imaging and Single Cell Expression from Tumor Bulk and Corpus Callosum Reveal Glioma Stem Cells with Unique Regulatory Programs"

_cancers, 2025, doi:10.3390/cancers17233851_

Round 1

Reviewer 1 Report

Comments and Suggestions for Authors

In my eyes, the manuscript offers important cues to the pathogenesis of the glioma recurrence and deserves publishing. However, if expanded to include more genes, the described CC-Iv glioma cell transcriptional signature characterising a more aggressive cancer cell phenotype, will probably be transformed into the mesenchymal phenotype. Just for the authors convenience: A while ago we demonstrated that mesenchymal stromal cells injected into the contralateral striatum of rats with experimental stroke entered the Corpus callosum and moved on towards the other hemisphere (Namestnikova et al. Therapeutic Efficacy and Migration of Mesenchymal Stem Cells after Intracerebral Transplantation in Rats with Experimental Ischemic Stroke. Bull Exp Biol Med 175, 116–125 (2023). https://doi.org/10.1007/s10517-023-05822-1)

Reviewer 2 Report

Comments and Suggestions for Authors

This manuscript by Costa et al. presents a technically sophisticated and conceptually strong study that combines orthotopic xenografts, tissue clearing (CLARITY), light-sheet microscopy, and single-cell transcriptional profiling to dissect region-specific invasion mechanisms of glioma stem-like cells (GSCs). The authors identify a corpus callosum–specific invasion signature (CC-Iv) comprising seven genes (NES, CCND1, GUSB, NOTCH1, E2F1, EGFR, TGFB1), revealing spatial heterogeneity within the invasive front. The integration of in vivo imaging and transcriptomic resolution is particularly commendable and offers translational relevance.

The manuscript stands out for its methodological rigor and the spatial resolution achieved in mapping GSC infiltration patterns. The use of whole-brain clearing and light-sheet imaging provides an impressive anatomical context rarely achieved in glioma models. Moreover, the identification of a CC-specific transcriptional profile contributes to the growing understanding of how white matter microenvironments shape tumor behavior and resistance.

Major Comment

While the study compellingly characterizes invasive programs, it is important to understand whether such transcriptional profiles are static or plastic in response to therapy.

  • Have the authors considered applying temozolomide or another alkylating agent to their in vivo or in vitro model to assess whether the CC-Iv signature undergoes a phenotypic transition (e.g., from proneural toward mesenchymal) following treatment?
    Such an analysis would not only validate the functional relevance of the identified genes but also connect spatial heterogeneity with therapy-induced adaptation — an aspect increasingly recognized in glioma biology.

Minor Comment

The discussion could further benefit from a mechanistic perspective linking electrophysiological and ionic remodeling to stemness and invasive behavior in glioma. In recent years, several studies have demonstrated that changes in voltage-gated sodium (Nav1.1) currents and Na⁺/Ca²⁺ exchange activity can influence glioblastoma cell migration, stemness, and chemosensitivity, even driving differentiation programs that sensitize cells to alkylating agents. These findings may offer an interesting parallel to the authors’ observation of distinct transcriptional states between tumor core and white matter–infiltrating GSCs and ion channels might be a relevant class of transcripts significantly different in their different sets of mRNA.

Reviewer 3 Report

Comments and Suggestions for Authors

This study utilizes single-cell transcriptomics to explore invasion dynamics in glioma stem cells (GSCs) using an orthotopic mouse model. The authors identify a seven-gene corpus callosum invasion signature (CC-Iv) enriched in GSCs infiltrating white matter tracts. The study further suggests that this CC-Iv signature correlates with poor survival in low-grade glioma (LGG) patients, and identifies NOTCH1 as a central regulator of stemness and invasion. This study presents a technically elegant approach to dissecting spatial GSC heterogeneity, but the manuscript overstates causal claims and lacks sufficient validation and statistical rigor. Several concerns limit the strength of the conclusions:

  1. Limited generalizability due to single GSC line usage
    The entire analysis relies on a single patient-derived GSC line (GSC23), preselected for corpus callosum tropism. This raises concerns about the generalizability of the CC-Iv signature. Validation across multiple, molecularly diverse GSC lines is essential to support claims of a conserved white matter invasion program.

  2. Overstated conclusions regarding causality and function
    No functional validation of CC-Iv genes is provided. Claims about their roles in invasion, stemness, or migration are inferred from published literature, not demonstrated through assays such as migration, apoptosis, or neurosphere formation comparing CC-Iv high vs. low cells. The manuscript would be strengthened by either removing these functional claims or incorporating targetedloss- or gain-of-function experiments.

  3. Insufficient discussion of model and technical limitations
  • The use ofBALBc Nude mice, which lack adaptive immunity, is a critical factor when interpreting transcriptomic changes related to immune signaling. The rationale of using this model and whether this may indoroduce artifacts is not fully addressed. Also, while immune-related genes are discussed, the model's inability to recapitulate immune–tumor interactions is not adequately acknowledged.
  • Include more detailed metadata on the 164 cells analyzed (e.g., coverage per cell, gene dropout rates).
  • The 48-gene panel is derived from a larger pool of over 1,000 GSC-expressed genes (Lines 362–363), yet the curation process is not described in detail. This raises concerns aboutselection bias and potential omission of relevant genes.
  • Differential expression analyses are conducted on 48 genes, but p-values are reported withoutmultiple hypothesis testing correction. False discovery rate (FDR) or Bonferroni adjustment is needed to avoid inflated Type I error.
  • The fixed 6-week timepoint may capture a mid-stage invasion profile, but the manuscript does not clarify how this timing was selected or how it may influence transcriptomic signatures.
  • The authors state that the CC-Iv signature may be "constitutively elevated" in high-grade gliomas, thus limiting its prognostic utility. However,no data are shown comparing absolute CC-Iv expression levels between LGG and HGG to substantiate this interpretation.
  • To assess the robustness and conservation of the CC-Iv signature, the authors should analyze these genes in published glioma single-cell RNA-seq datasets. Such cross-validation would significantly strengthen the claim that CC-Iv defines a conserved invasive GSC subpopulation.
